# ARCHLOCK: LOCKING DNN TRANSFERABILITY AT THE ARCHITECTURE LEVEL WITH A ZERO-COST BINARY PREDICTOR

**Tong Zhou[1], Shaolei Ren[2], Xiaolin Xu[1]**
[1]Northeastern University, Boston, MA, USA    [2]UC Riverside, Riverside, CA, USA
{zhou.tong1, x.xu}@northeastern.edu, shaolei@ucr.edu

## ABSTRACT

Deep neural network (DNN) models, despite their impressive performance, are vulnerable to exploitation by attackers who attempt to transfer them to other tasks for their own benefit. Current defense strategies mainly address this vulnerability at the model parameter level, leaving the potential of architectural-level defense largely unexplored. This paper, for the first time, addresses the issue of model protection by reducing transferability at the architecture level. Specifically, we present a novel neural architecture search (NAS)-enabled algorithm that employs zero-cost proxies and evolutionary search, to explore model architectures with low transferability. Our method, namely ArchLock, aims to achieve high performance on the source task, while degrading the performance on potential target tasks, i.e., locking the transferability of a DNN model. To achieve efficient cross-task search without accurately knowing the training data owned by the attackers, we utilize zero-cost proxies to speed up architecture evaluation and simulate potential target task embeddings to assist cross-task search with a binary performance predictor. Extensive experiments on NAS-Bench-201 and TransNAS-Bench-101 demonstrate that ArchLock reduces transferability by up to 30% and 50%, respectively, with negligible performance degradation on source tasks (<2%). The code is available at https://github.com/Tongzhou0101/ArchLock.

## 1 INTRODUCTION

It is a common practice today to transfer a pre-trained deep neural network (DNN) from one application domain to another one, as demonstrated in Guo et al. (2019); Jiang et al. (2022). These advantages, however, also provide an incentive for attackers to illicitly exploit well-trained models by transferring them to their desired tasks, resulting in a violation of model owners' rights. To mitigate the vulnerability of unauthorized model transfer, prior work restricts model usage by introducing a training strategy to optimize the weights Wang et al. (2022). Nonetheless, such a weight-level modification cannot be preserved once attackers fine-tune this model to the domain of interest with enough training data, since they will further adjust the model weights to improve the transferability.

In contrast, we focus on achieving protection at the architecture level, recognizing that the neural architecture fundamentally determines the model accuracy Zoph & Le (2017) and hence plays a critical role in model transferability Kornblith et al. (2019); Zhou et al. (2021). Despite the critical significance of architectural considerations in model transferability, previous mitigation efforts have largely overlooked this aspect. Therefore, our work seeks to bridge this gap by addressing the question: **how to mitigate the risk of unauthorized transfer by reducing transferability at the architecture level?** In particular, we expect the DNN to perform well on the source task while showing limited transferability to other tasks.

Although our proposed defense philosophy is intuitive, manually designing an architecture with low transferability is challenging, due to the large design space and requirements for domain knowledge of architecture characteristics associated with transferability. Consequently, neural network search (NAS) is envisioned as a promising solution due to its effectiveness in architecture design Liu et al. (2018b); Real et al. (2019); Zhou et al. (2022). Even so, how to utilize NAS to reduce DNN

transferability for model protection has remained an open problem. Indeed, it is significantly more challenging than improving DNN transferability Ding et al. (2022); Pasunuru & Bansal (2019), since the target task for the latter is well-specified but not for our case. Therefore, in addition to the common challenges faced by existing NAS algorithms, i.e., how to efficiently evaluate the performance of architecture in the search space, we have to address additional challenges, including **(a)** how to determine and simulate the potential target task if it is not specified; **(b)** how to address performance evaluation for the simulated task; and **(c)** how to preserve the performance on the source task but degrade performance on the target task.

In this paper, we formulate the transferability reduction problem as a cross-task search NAS problem and propose a novel algorithm, called ArchLock. Specifically, for challenge **(a)**, we simulate the target task by generating its task embedding based on the source task with a specified similarity, since the target task should be similar to the source task, so as to benefit (i.e., obtain the performance gain) from transfer learning (Sec. 3.1). For challenge **(b)**, we meta-train a binary predictor with zero-cost proxies to indicate the relative performance, which can achieve efficient architecture evaluation and generalize to unseen tasks (Sec. 3.2). Last, we design a cross-task search algorithm using evolutionary search with a rank-based fitness score to guide the search (Sec. 3.3). Overall, our contributions can be summarized as follows:

• This work is the first to mitigate unauthorized transfer at the architecture level, where the searched architecture excels on the source task but exhibits degraded performance on the target task, regardless of the amount of data available to the attacker.

• We develop a binary predictor using multiple zero-cost proxies to accelerate NAS. The predictor incorporates task characteristics as an additional input, enabling efficient architecture evaluation and generalization to unseen tasks, thus assisting in cross-task search with a rank-based fitness score.

• Through extensive experiments on NAS-Bench-201 and TransNAS-Bench-101, we demonstrate that ArchLock can reduce transferability by up to about 30% and 50%, respectively, while maintaining negligible performance degradation on the source task (<2%).

## 2 RELATED WORK

### 2.1 EFFECT OF ARCHITECTURE ON MODEL TRANSFERABILITY

Previous works have demonstrated that the architecture design affects the model transferability Zhou et al. (2021); Kornblith et al. (2019). For example, Zhou *et al.* demonstrate that the Transformer-based architectures can be better transferred to 13 tasks than ConvNets Zhou et al. (2021). Besides, Zoph *et al.* search well-performed cells only on CIFAR-10 and transfer them to ImageNet with good performance. However, Dong & Yang (2020) shows that directly transferring an optimal architecture from a source task to a target task might not always yield good performance, since the rank correlation of architecture performances on different tasks is not perfectly positive. This work suggests that a single-task search cannot ensure that the top-1 architecture on the source task also has the best performance on the target task. Therefore, some cross-task search NAS algorithms are proposed. For example, Ding et al. (2022) builds a predictor for each task, then accumulates the gradient from these predictors to guide the architecture search. Besides, Pasunuru & Bansal (2019) utilizes a controller to obtain a joint reward to achieve a multi-task search.

However, these existing cross-task NAS algorithms aim to find the architecture performing well on several specified tasks. In sharp contrast, our proposed design only expects generating model architectures that perform well on the source task with reduced transferability to uncertain target tasks, which raises challenges including achieving efficient architecture performance evaluation and estimating performance on unseen tasks.

### 2.2 ACCELERATION FOR ARCHITECTURE EVALUATION

Evaluating an architecture performance by training it until convergence is computationally expensive. To overcome this limitation, some works design the performance predictors to accelerate NAS, which are usually built as regression models that are trained using architecture encodings and their corresponding validation performances (e.g., accuracy for classification problems) Liu et al. (2018a);

Shi et al. (2020). However, collecting these training data is also expensive and the predictor may not have a high correlation between the predicted result with the actual performance White et al. (2021). As an improvement, Dudziak *et al*. propose a predictor using the binary relationship to rank architectures, which requires fewer training samples but has a higher correlation Dudziak et al. (2020).

In contrast to the above methods that still require training architectures, several theoretical metrics have been proposed recently that can measure the performance of an architecture at the initialization stage Lee et al. (2019a); Mellor et al. (2021); Lin et al. (2021), thus greatly reducing the computation burden in NAS. Since the computation cost of such a metric is negligible (e.g., less than 5 seconds Krishnakumar et al. (2022)), they are used as zero-cost proxies (ZCPs) in previous NAS works Chen et al.. However, a single ZCP cannot provide accurate rank or generalize well among various tasks Chen et al.; Abdelfattah et al. (2021). Therefore, Krishnakumar et al. (2022) incorporates 13 proxies into the surrogate models used by NAS algorithms, showing up to 42% performance improvement. However, we leverage ZCPs in a different way, with details in Sec. 3.2.

### 2.3 NAS PREDICTORS WITH META-LEARNING

Given that explicitly measuring the performance of architectures on simulated target tasks is impossible, we draw inspiration from the following works to leverage meta-learning in training our proposed predictor. In particular, recent NAS algorithms adopt meta-learning to improve its generalizability on other novel tasks Elsken et al. (2020); Lee et al. (2021a;b). For example, MetaNAS optimizes both the meta-architecture and the meta-weights during meta-training, allowing adaptation to unseen tasks during testing Elsken et al. (2020). Besides, Lee *et al*. propose MetaD2A to meta-train a performance predictor on the ImageNet classification task to accelerate the search Lee et al. (2021a), which generalizes well on other classification tasks and reduces the search cost from O(N) to O(1) for multi-task search. Therefore, we also adopt meta-learning in our predictor design, thus being able to estimate the architecture performance on simulated tasks. Moreover, we extend the applicability of the meta-trained predictor beyond the classification task.

## 3 OUR PROPOSED FRAMEWORK: ARCHLOCK

Our goal is to utilize NAS to find an architecture that can mitigate unauthorized transferability. Specifically, except for achieving the optimal performance on the single task as conventional NAS, we expect the searched architecture can restrain the model usage to the source task, i.e., perform well on the source task, but hard to transfer to the target task with good performance. To this end, we propose ArchLock, which learns the cross-task performance of architectures via an evolutionary search. Specifically, ArchLock uses a rank-based fitness score to guide the search. Further, we introduce a binary predictor using zero-cost proxies to rank the architectures in ArchLock, where the pairwise relationship will be task-dependent, thus we can simulate the potential task embedding to evaluate the architecture performance on uncertain tasks.

**Problem Formulation:** Conventional NAS algorithms focus on finding an optimal architecture that achieves high performance on a single task (denoted as $\mathcal{S}$), while satisfying the hardware constraints, e.g., memory consumption and/or inference latency Zoph & Le (2017); Chen et al., which can be formulated as a bi-level optimization problem:

$$a' = \arg\min_{a \in \mathcal{A}} \mathcal{L}_{\mathcal{S}}(f_a(X_{\mathcal{S}}^{val}, W^*); Y_{\mathcal{S}}^{val}), \quad s.t. \ W^* = \arg\min_{W} \mathcal{L}_{\mathcal{S}}(f_a(X_{\mathcal{S}}^{tr}, W); Y_{\mathcal{S}}^{tr}), \quad (1)$$

where $(X_{\mathcal{S}}^{tr}, Y_{\mathcal{S}}^{tr})$ and $(X_{\mathcal{S}}^{val}, Y_{\mathcal{S}}^{val})$ are the training data and validation data of $\mathcal{S}$, respectively, and $\mathcal{L}_{\mathcal{S}}$ is the loss function of $\mathcal{S}$. We use $\mathcal{A}$ to denote the search space and $f_a$ to represent the network associated with the architecture $a$, where $a$ is an architecture candidate in $\mathcal{A}$ and $W$ is the corresponding weights trained for $\mathcal{S}$.

However, in order to mitigate the vulnerability of unauthorized transfer, we have to consider the architecture performance on potential target tasks, which is denoted as $\mathcal{T}_i$. Similarly, this problem can be formulated as a bi-level optimization problem, but we aim to minimize the performance of the network on $\mathcal{T}_i$, while maximizing its performance on $\mathcal{S}$, i.e., the specific source task. The optimal

architecture $a^*$ can be found by solving:

$$a^* = \underset{a \in \mathcal{A}}{\arg\min} \, \mathcal{L}_{\mathcal{S}}(f_a(X_{\mathcal{S}}^{val}, W^*); Y_{\mathcal{S}}^{val}) - \beta \sum_{\mathcal{T}_i} \mathcal{L}_{\mathcal{T}_i}(f_a(X_{\mathcal{T}_i}^{val}, W_{\mathcal{T}_i}^*); Y_{\mathcal{T}_i}^{val}),$$

$$s.t. \, W_{\mathcal{T}_i}^* = \underset{W}{\arg\min} \, \mathcal{L}_{\mathcal{T}_i}(f_a(X_{tr}^{\mathcal{T}_i}, W); Y_{tr}^{\mathcal{T}_i}) \tag{2}$$

where $(X_{\mathcal{T}_i}, Y_{\mathcal{T}_i})$ is the labeled pair from the task $\mathcal{T}_i$ and constraints in Eq. (1) still apply. Also, $\beta$ controls the balance between performance maximization on the source task and minimization on the target task. Especially, when $\beta$ equals 0, Eq. (2) will be degraded to Eq. (1), thus we can get the performance of NAS without the security concern of transferability reduction as the baseline.

However, solving Eq. (2) faces three challenging problems: 1) how to simulate the task domain if it is unknown; 2) how to efficiently measure the architecture performance and let it generalize to unseen tasks as well, and 3) how to achieve cross-task search. We tackle these challenges in Sec. 3.1, Sec. 3.2, and Sec. 3.3, respectively.

## 3.1 TASK EMBEDDING

It is worth noting that fully protecting the model by reducing the model transferability to *all* potential target tasks is practically infeasible, i.e., one could not pre-know the tasks or applications of interest to the adversary, which is also not our target in this work. This is also true for other types of defenses, e.g., adversarial training can only robustify the DNN model subject to an input perturbation bound Wang et al. (2019). Here, we expect the transferability to some tasks can degrade so that we can defend against partial potential attacks.

In general, simulating the potential target tasks by generating their actual dataset can be difficult if not impossible, since we need to know exactly the attacker's tasks. To address this challenge, we aim to manipulate the feature space directly to save the computational cost of dataset generalization, i.e., generate the possible target task embedding. The bright side is that the target task in general is close to the source task in order to benefit from transfer learning. In this case, we can simulate some potential target tasks during the search to assist cross-task search when the target task is unknown.

To capture the characteristics of a task as an embedding vector, we leverage the Fisher Information matrix (FIM) similar to Achille et al. (2019) and Huang et al. (2022), which can be expressed by:

$$F = \mathbb{E}_{X,Y \sim \hat{p}(X)p_w(Y|X)} \left[ \nabla_w \log p_w(Y|X) \nabla_w \log p_w(Y|X)^T \right], \tag{3}$$

where $\hat{p}$ is the empirical distribution defined by the dataset including the input $X$ and the label $Y$. Specifically, the modern DNN model can be seen as two parts: a feature extractor and a task-dependent head, e.g., a classifier for the classification task. Here, we use ResNet-50 pre-trained on ImageNet as the feature extractor and train the head for any given task. After training, we compute the FIM for the feature extractor with the same approximation in Achille et al. (2019) to represent the task as a fixed dimensional vector $\mathbf{z}$.

With the source task embedding $\mathbf{z}_{\mathcal{S}}$ obtained using the above method, we can simulate the task embedding $\mathbf{z}_{\mathcal{T}_i}$ of potential target tasks $\mathcal{T}_i$. Here we use the cosine similarity for measurement, where the value is between [0,1] and the larger value indicates higher similarity. Therefore, given $\mathbf{z}_{\mathcal{S}}$ and desired cosine similarity $d$, we can generate $\mathbf{z}_{\mathcal{T}_i}$ following :

$$\mathbf{z}_{\mathcal{T}_i} = d \cdot \mathbf{z}_{\mathcal{S}} + \sin(\arccos(d)) \cdot e_i \cdot ||\mathbf{z}_{\mathcal{S}}||, \tag{4}$$

where $e_i$ is orthogonal to $\mathbf{z}_{\mathcal{S}}$ and can be obtained via Gram–Schmidt process. The derivation of Eq. (8) is shown in Appendix A. Besides, the orthogonal vector will vary for different $\mathbf{z}_{\mathcal{T}_i}$.

The advantage of using such a method is that we can simulate possible target tasks without explicitly generating the samples, and the latent space manipulation is more representative. With the simulated task embedding, we expect that they can represent or approach some real target tasks. Since we do not have the ground truth label for the simulated task, we cannot estimate the architecture performance using supervised learning. Instead, we use the task embedding as an input of the predictor introduced below, where the predictor will be meta-trained to be generalized to the unseen task embedding.

## 3.2 ZERO-COST BINARY PREDICTOR

One challenge we face in developing ArchLock is estimating the architecture's performance on unseen tasks. Inspired by the success of meta-learning in generalizing models to unseen tasks Finn et al. (2017); Lee et al. (2021a), we propose to train a predictor that can adapt to simulated tasks following the procedure in MetaD2A Lee et al. (2021a). However, unlike the predictor in MetaD2A that uses validation performance as the label for meta-training, we use zero-cost metrics as proxies to measure architecture performance at initialization, thus avoiding architecture training and saving computational costs.

While various zero-cost proxies have been used in NAS to accelerate the search process, relying on a single proxy as a measure of architecture's performance is unreliable White et al. (2021). The searched architecture may not perform well, and the zero-cost proxy (ZCP) itself may not generalize to other tasks. To address this, we incorporate multiple data-dependent ZCPs to obtain a more accurate estimation, including *fisher*, *flops*, *grad-norm*, *grasp*, *jacov*, *nwot*, and *snip*, as summarized in Krishnakumar et al. (2022). These proxies provide complementary information about the characteristics of an architecture's performance. The details of these proxies and the rationale for their selection are discussed in Appendix B.

Our objective is to obtain architecture rankings rather than precise performance values. Therefore, we propose a binary predictor $P$ that learns pairwise relations between two architectures instead of predicting ordinal regression. Such a design can improve the overall ranking accuracy of the predictor since learning a relative performance for two architectures will be easier than predicting accurate rankings for multiple architectures. More importantly, we incorporate task characteristics into the predictor using task embedding, so it can be used in the cross-task search to efficiently measure the architecture performance given different task embeddings.

Overall, the inference process of the proposed predictor is shown in Fig. 1. The graph encoder converts the directed acyclic graph of the architecture cell into an adjacency matrix, which is then flattened into a vector. As for the binary predictor, it is built with a multi-layer perceptron with a softmax activation function and takes the concatenated architecture encoding pair and the pre-computed task embedding as inputs. During training, the ground-truth label is calculated based on the ZCPs proxies using an ensemble method. Specifically, the label is set to 1 if more proxies indicate that the first architecture outperforms the second in an architecture pair. The design and training details are described in Sec. 4.3. During inference, the predictor outputs two logits $p_1$ and $p_2$. If $p_1 > p_2$, it indicates architecture $a_1$ outperforms $a_2$ on the task associated with task embedding $z$. In summary, the

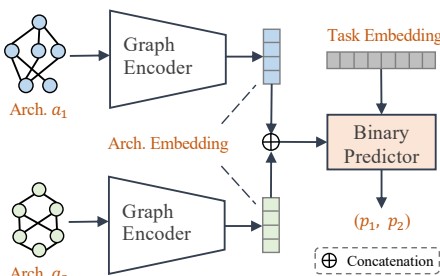

Figure 1: The inference process of our proposed predictor, where the inputs include the embedding of a pair of architectures along with the pre-computed task embedding. $p_1 > p_2$ indicates that $a_1$ outperforms $a_2$ for the given task.

proposed predictor has the following properties: (1) It is scalable to any size of the architecture pool when combined with the sorting algorithm; (2) It uses zero-cost proxies as supervision, eliminating the need to train several architectures from scratch and reducing training costs; (3) It incorporates multiple ZCPs to obtain complementary information, mitigating the bias introduced by a single proxy; (4) It takes both architecture and task embeddings into account, leveraging meta-learning for training and enabling generalization to unseen tasks.

## 3.3 CROSS-TASK EVOLUTIONARY SEARCH

In ArchLock, we utilize evolutionary search due to its computational efficiency compared to RL-based NAS Zoph & Le (2017). Also, unlike pruning-based approaches Chen et al., its mutation mechanism can help avoid local minima, allowing for a more thorough exploration of the solution space.

The core algorithm of cross-task search in ArchLock is shown in Alg. 1. To achieve the transferability reduction via a cross-task search, we use the meta-trained predictor along with a sorting algorithm, i.e., topological sorting in this case, to rank the performance of architectures for $\mathcal{S}$ and $\mathcal{T}_i$, denoted as BPwSort in Alg. 1. In particular, we incorporate the rankings for all $\mathcal{T}_i$ as one using majority voting,

denoted as $\mathcal{T}_{rank}$. We can then use the rankings of architecture on $\mathcal{S}$ and $\mathcal{T}$ to design a fitness score as guidance for the cross-task search. The rank-based fitness score f is designed as:

$$\text{f} = \mathcal{T}_{rank} - \lambda * \mathcal{S}_{rank}, \tag{5}$$

where $\mathcal{S}_{rank}$ denotes the architecture rankings on $\mathcal{S}$ and we set $\lambda$ to 2 to assign the source task higher weight. In this case, we are able to find an architecture that satisfies Eq. 2, i.e., the performance will be degraded on the target task while being well on the source task compared to the conventional NAS. The functions BPwSort and UpdateHistory are shown in Appendix C.

## 4 EXPERIMENTS

We evaluate ArchLock on two state-of-the-art NAS benchmarks: NAS-Bench-201 (NB-201) Dong & Yang (2020) and TransNAS-Bench-101 (TNB-101) Duan et al. (2021). These are the only two accessible options that consist of multiple tasks. Performance evaluation is based on the validation performance provided by the benchmarks for fair comparison. We compare three search schemes: one is source-only search and the other two are cross-task search, with details described in Sec. 4.2.

### 4.1 SEARCH SPACE AND TASKS

**NAS-Bench-201** is a popular NAS benchmark dataset with cell-based search space. Each cell has 4 nodes and 6 edges, where each edge represents an operation selected from 5 options, i.e., zeroize, skip connection, 1×1 convolution, 3×3 convolution, and 3×3 average pooling. It includes over 15,000 neural network architectures across three image classification tasks, including CIFAR-10, CIFAR-100, and ImageNet-16-120 (i.e., 16×16 input size with 120 classes).

**TransNAS-Bench-101** is proposed to facilitate the cross-task NAS by including the architecture performance across 7 different tasks on the same dataset, including scene classification (**SC**), object classification (**OC**), room layout (**RL**), jigsaw (**JS**), semantic segmentation (**SS**), surface normal (**SN**), and autoencoding(**AE**), where the labels for each task collected from Taskonomy Zamir et al. (2018). It provides two types of search space, i.e., macro-based and cell-based, where we experiment on the latter with more than 4k architectures. Each cell consists of 4 nodes and 6 edges, where the operations for each edge include zeroize, skip connection, 1×1 convolution, and 3×3 convolution.

---

**Algorithm 1** Cross-task Search in ArchLock

---

**Require**: Predictor $P$, Task embeddings $\mathbf{z}_{\mathcal{S}}$, $\mathbf{z}_{\mathcal{T}_i}$
**Parameters**: Population size $n$, Sample size $m$, History size $k$

1: $popu \leftarrow \emptyset, history \leftarrow \emptyset$
2: **procedure** NEWEPOCH($epoch, P, \mathbf{z}_{\mathcal{S}}, \mathbf{z}_{\mathcal{T}_i}$)
3:     **if** $epoch < n$ **then**
4:         $arch \leftarrow$ Randomly sample from search space
5:         $popu$.append($arch$)
6:         UpdateHistory($history, arch, k, P, \mathbf{z}_{\mathcal{S}}, \mathbf{z}_{\mathcal{T}_i}$)
7:     **else**
8:         $\mathcal{S}_{rank}, \mathcal{T}_{rank} \leftarrow$ BPwSort($popu, P, \mathbf{z}_{\mathcal{S}}, \mathbf{z}_{\mathcal{T}_i}$)
9:         Calculate f for $arch \in popu$ (Eq.(5))
10:        $samples \leftarrow$ Randomly sample $m$ from $popu$
11:        $parent \leftarrow arch$ with highest f in $samples$
12:        $child \leftarrow$ Mutate($parent$)
13:        $popu$.popleft()
14:        $popu$.append($child$);
15:        UpdateHistory($history, child, k, P, \mathbf{z}_{\mathcal{S}}, \mathbf{z}_{\mathcal{T}_i}$)
16:     **end if**
17: **end procedure**
18: $a* \leftarrow arch$ with highest f in $history$

---

### 4.2 DIFFERENT SEARCH SCHEMES

**ArchLock-S**: This is a source-only search scheme, which only conducts ArchLock on the source task $\mathcal{S}$, i.e., setting $\beta$ in Eq. (2) to 0. After getting the architecture, we will evaluate how it performs when trained on the target task. It is a baseline as the comparison with cross-task search schemes.

**ArchLock-TU**: It conducts cross-task NAS in the case that the target task is unknown (TU). We use simulated tasks with embedding generated based on the method introduced in Sec. 3.1. We measure the performance of the searched architecture on the target task, which is unseen during the search.

**ArchLock-TK**: It conducts cross-task NAS in the case that the target task is known (TK), where the objective follows Eq. (2) with only one $\mathcal{T}$ obtained from the certain target task.

Table 1: The transferability of searched architecture on NB-201 with different search schemes. We report both the Top-1 validation accuracy, i.e., *Acc. (%)*, and rank percentile, i.e., *Pct. (%)* to evaluate the performance of architectures. We use *APT* to denote the average rank percentile on target tasks.

| Source / Target | Method | CIFAR-10 | | CIFAR-100 | | ImageNet-16-120 | | *APT* |
|---|---|---|---|---|---|---|---|---|
| | | Acc. | *Pct.* | Acc. | *Pct.* | Acc. | *Pct.* | |
| CIFAR-10 | ArchLock-S | 91.40 | 99.99 | 72.96 | 99.95 | 46.77 | 99.94 | 99.95 |
| | ArchLock-TU | 90.41 | 97.93 | 70.24 | 95.81 | 44.11 | 95.44 | 95.62 |
| | ArchLock-TK | 90.55 | 98.56 | 68.67 | 82.78 | 41.80 | 84.07 | **83.37** |
| CIFAR-100 | ArchLock-S | 91.18 | 99.91 | 73.25 | 99.99 | 46.5 | 99.84 | 99.88 |
| | ArchLock-TU | 89.69 | 93.45 | 71.18 | 98.98 | 41.47 | 82.11 | 87.78 |
| | ArchLock-TK | 89.07 | 84.47 | 71.23 | 99.07 | 37.27 | 54.84 | **68.65** |
| ImageNet-16-120 | ArchLock-S | 91.23 | 99.94 | 72.74 | 99.92 | 47.27 | 99.99 | 99.93 |
| | ArchLock-TU | 89.53 | 91.95 | 69.16 | 87.60 | 45.22 | 98.52 | 89.77 |
| | ArchLock-TK | 88.73 | 76.59 | 67.69 | 72.72 | 45.58 | 99.25 | **74.65** |

## 4.3 EXPERIMENTAL SETUP

**Meta-trained predictor**: we build a binary predictor using a multi-layer perceptron with four layers. To train the predictor, we use ImageNet to build our training set as Lee et al. (2021a). Specifically, the training set includes architecture pairs randomly sampled from Nas-Bench-201, the task embedding of the tasks associated with the sub-set of ImageNet, and the binary label decided by the values of ZPCs. Specifically, if the majority of ZPCs indicate the first architecture outperforms the second, then the label is 1; otherwise, the label would be 0. Following the same meta-training procedure as Lee et al. (2021a), our predictor can generalize to unseen tasks, where the task similarity will affect the performance of the meta-trained predictor as discussed in Sec. 5.2. Our experiments are conducted with the PyTorch framework using NASLib library Ruchte et al. (2020).

**Simulated task embeddings:** For the following experiments, we simulate 10 task embeddings $\mathcal{T}_i$, where the cosine similarity of $\mathcal{S}$ and $\mathcal{T}_i$ is set to 0.9. These settings will be the same unless specified otherwise, and we will discuss how they affect the search if the values are changed in Sec. 5.2.

**Measurement:** For classification tasks, we use top-1 validation accuracy as a metric to measure the performance. Besides, we also use rank percentile, denoted as *Pct.*, to measure how a candidate performs among all architectures in the search space, since the architecture ranking matters in NAS. For example, if *Pct.* of an architecture is 90%, it means that this architecture performs better than 90% of all architectures in the search space. In order to measure the performance of three search schemes, we also introduce the average rank percentile on target tasks, denoted as *APT*, which takes the *Pct.* on all tasks except the one same as the source task.

## 4.4 RESULTS ON NAS-BENCH-201

We apply three search schemes on the NB-201 benchmark with 3 datasets and take the average of results from several runs for each experiment as reported in Tab. 1. From this table, we can see that ArchLock-S can achieve high performance (with *Pct.* >99%) on target tasks, even though it only searches on the source task, which indicates the searched architectures in this case are more vulnerable to unauthorized transfer. The reason is that the correlation on each task pair over architectures in NB-201 is high, with details shown in Appendix D.

However, if we change the search scheme to ArchLock-TU, the *APT* will have an obvious drop, with up to 12% when CIFAR-100 is the source task, while the *Pct.* on the source task does not drop much, i.e., < 2%. The rationale for employing *Pct.* as a descriptor of performance lies in the insufficiency of accuracy alone to fully represent the architectural performance across the entire search space. Specifically, since the architecture performance does not follow the uniform distribution in the search space, i.e., a small accuracy drop will significantly change the architecture ranking, making the result no longer attractive to attackers. Moreover, ArchLock-TK shows the highest *APT* drop since we can directly use the specified target task in cross-task search, achieving the maximal transferability reduction. Also, it can preserve the performance of the source task, which is close to the source-only search ArchLock-S.

Table 2: The transferability of searched architecture on TNB-101 with different search schemes. We use *APT* to denote the average rank percentile on target tasks.

| Source / Target | Method | SC | OC | RL | JS | SS | SN | AE | APT |
|---|---|---|---|---|---|---|---|---|---|
| **SC** | ArchLock-S | 99.99 | 99.83 | 59.91 | 89.71 | 75.88 | 99.32 | 95.97 | 86.77 |
| | ArchLock-TU | 98.96 | 66.36 | 59.17 | 67.37 | 75.89 | 67.47 | 44.96 | 63.53 |
| | ArchLock-TK | 99.73 | 39.48 | 34.03 | 40.43 | 46.67 | 54.49 | 30.19 | **40.88** |
| **OC** | ArchLock-S | 89.21 | 99.93 | 73.61 | 63.87 | 69.92 | 82.10 | 26.76 | 67.58 |
| | ArchLock-TU | 76.94 | 97.10 | 73.77 | 41.41 | 63.10 | 62.33 | 63.21 | 63.46 |
| | ArchLock-TK | 47.96 | 99.65 | 59.91 | 38.95 | 42.04 | 35.24 | 26.76 | **41.81** |
| **RL** | ArchLock-S | 98.47 | 94.85 | 99.78 | 83.26 | 74.34 | 99.85 | 77.02 | 87.97 |
| | ArchLock-TU | 49.38 | 67.91 | 97.74 | 60.54 | 46.55 | 64.02 | 49.62 | 56.34 |
| | ArchLock-TK | 37.76 | 40.56 | 99.75 | 39.49 | 45.46 | 22.95 | | **36.38** |
| **JS** | ArchLock-S | 97.98 | 97.77 | 65.04 | 99.68 | 87.82 | 97.58 | 72.31 | 86.42 |
| | ArchLock-TU | 67.51 | 61.06 | 76.90 | 98.52 | 67.54 | 53.17 | 37.38 | 60.59 |
| | ArchLock-TK | 35.46 | 31.91 | 42.46 | 99.58 | 54.24 | 32.08 | 19.67 | **35.97** |
| **SS** | ArchLock-S | 81.81 | 61.65 | 36.91 | 97.35 | 99.87 | 63.40 | 28.03 | 61.53 |
| | ArchLock-TU | 68.72 | 38.43 | 34.59 | 47.48 | 98.47 | 85.06 | 19.21 | 48.91 |
| | ArchLock-TK | 40.84 | 30.98 | 30.07 | 30.42 | 99.34 | 46.36 | 11.33 | **31.67** |
| **SN** | ArchLock-S | 97.02 | 97.13 | 79.05 | 73.19 | 76.17 | 99.97 | 81.49 | 84.00 |
| | ArchLock-TU | 67.95 | 54.89 | 56.14 | 50.95 | 70.45 | 98.44 | 52.44 | 58.80 |
| | ArchLock-TK | 38.95 | 33.17 | 31.00 | 37.74 | 43.45 | 99.61 | 15.21 | **33.25** |
| **AE** | ArchLock-S | 37.37 | 62.37 | 84.92 | 55.26 | 34.89 | 74.40 | 99.83 | 58.22 |
| | ArchLock-TU | 36.33 | 48.29 | 54.81 | 50.08 | 31.89 | 46.84 | 98.08 | 44.71 |
| | ArchLock-TK | 31.13 | 47.95 | 56.13 | 45.36 | 30.39 | 29.58 | 99.74 | **40.09** |

## 4.5 RESULTS ON TRANS-NAS-101

The performances of three search schemes on TNB-101 are shown in Tab. 2, where the architecture performance is reported as the average of multiple runs for each source-target pair. From Tab. 2, we can find that two cross-task search schemes, i.e., ArchLock-TU and ArchLock-TK, can still maintain good performance on the source task as the source-only search ArchLock-S, while the performance on the target tasks drops drastically. Besides, since the correlation of some task pairs over architectures in TNB-101 is low (see Appendix E), e.g., architecture with good performance on **OC** could be bad on **AE**, the *APT* is relatively low for ArchLock-S compared it on NB-201. The intuition is that the target task could benefit more if the source task does the same kind of vision task.

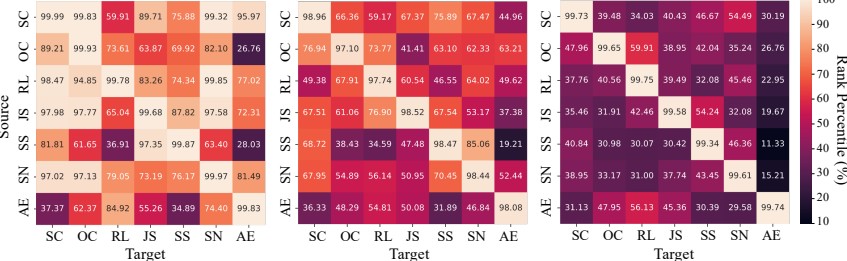

Figure 2: From left to right: ArchLock-S, ArchLock-TU, ArchLock-TK. The darker color indicates higher transferability reduction.

Although the *APT* of ArchLock-S is relatively low, we can still enhance the transferability reduction via cross-task search, where the results in Fig. 2 can show the performance difference among three search schemes. For example, when transferring **SS** to **SC**, ArchLock-S finds an architecture with *Pct.* of 81.81%, while ArchLock-TU and ArchLock-TK reduce the transferability even lower with *Pct.* of 68.72% and 40.48%, respectively. Besides, each heatmap reflects the asymmetric characteristics of transferability, i.e., transferring one task to another is different from the otherwise (e.g., 65.04% for **JS** to **RL** while 83.26% for **RL** to **JS** in ArchLock-S).

## 5 DISCUSSION

### 5.1 NAS ALGORITHMS

We compare ArchLock with SOTA source-only NAS algorithms on TNB-101, with **SC** denoting the source task and others representing the target tasks. The results are shown in Tab. 3. Note that even in this case the average Spearman rank correlation of all source-target task pairs is just ∼ 0.7, our

Table 3: Comparison among ArchLock and SOTA source-only NAS, including RS Bergstra & Bengio (2012), REA Real et al. (2019), BONAS Shi et al. (2020) and weakNAS Wu et al. (2021). The results are reported using the percentile (%) of architecture performance in the TNB-101 search space.

|  |  | Source-only | | | | | Cross-task | |
|---|---|---|---|---|---|---|---|---|
|  |  | RS | REA | BONAS | weakNAS | ArchLock-S | ArchLock-TU | ArchLock-TK |
| Source | SC | 99.68 | 99.85 | 99.34 | 99.71 | 99.99 | 98.96 | 99.73 |
|  | OC | 83.30 | 78.83 | 75.39 | 95.87 | 99.83 | 66.36 | 39.48 |
|  | RL | 60.86 | 62.30 | 63.31 | 76.95 | 59.91 | 59.17 | 34.03 |
| Target | JS | 78.78 | 84.72 | 64.82 | 76.95 | 89.71 | 67.34 | 40.43 |
|  | SS | 88.72 | 76.90 | 84.79 | 72.61 | 75.88 | 75.89 | 46.67 |
|  | SN | 97.00 | 77.34 | 86.79 | 98.14 | 99.32 | 67.47 | 54.49 |
|  | AE | 67.02 | 49.53 | 65.48 | 75.88 | 95.97 | 44.96 | 30.19 |
| APT | | 79.28 | 71.60 | 73.43 | 82.73 | 86.87 | **63.54** | **40.88** |

cross-task search schemes ArchLock-TU and ArchLock-TK can further reduce the *APT* to 63.54% and 40.88%, where *APT* for source-only NAS can be as high as over 80%. It demonstrates the effectiveness of the proposed cross-task search in mitigating unauthorized transfer.

## 5.2 SIMULATED TARGET TASK EMBEDDING

The simulated target task embedding plays an important role in ArchLock-TU. We experiment on NB-201 with CIFAR-100 as the source task and the other two as target tasks for investigation.

**The number of embeddings.** We experiment with different numbers of embeddings, i.e., 5, 10, and 15, respectively. From the results shown in Fig. 3, we can notice the gain in transferability reduction as the number of embeddings increases, i.e., the architecture performance does not change much on the source task but is degraded more on the target task with more simulated target task embeddings involved. The reason could be that the cross-task search covers more possible target tasks, which increases the possibility of the actual target task being included.

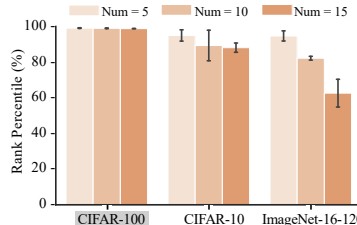

Figure 3: ArchLock-TU with various numbers of simulated embeddings.

Table 4: The performance of ArchLock-TU with various similarities of simulated task embeddings. The results are reported in terms of the rank percentile (%) of the architecture performance in NB-201 search space.

| Similarity $d$ | Source | Target | |
|---|---|---|---|
|  | CIFAR-100 | CIFAR-10 | ImageNet-16-120 |
| 0.3 | 99.07 | 94.34 | 97.56 |
| 0.5 | 98.89 | 95.82 | 92.59 |
| 0.9 | 98.98 | **93.45** | **82.11** |

**The similarity of embeddings.** We explore various cosine similarities between the source task embedding and the simulated target embedding, using $d$ values of 0.3, 0.5, and 0.9 (see Eq. 8). In Table 4, we find that ArchLock-TU exhibits higher transferability reduction as similarity increases from 0.3 to 0.9 (97.56% vs 82.11% for ImageNet-16-120). This is likely because simulating low-similarity tasks provides less informative guidance, potentially leading to a search towards less similar target tasks resulting in a lower transferability reduction for the actual target task.

## 6 CONCLUSION

This paper presents ArchLock, a cross-task NAS framework aimed at mitigating unauthorized DNN transfer. Our framework focuses on finding an architecture that performs well on the source task but exhibits degraded performance when transferred to a similar target task. To achieve this objective efficiently, we introduce a binary predictor that utilizes 7 zero-cost proxies to save computational costs. Additionally, we simulate the target task embeddings as an input of the predictor. By leveraging meta-learning, this predictor can be generalized to unseen tasks, thereby assisting cross-task search with a rank-based fitness score. Extensive experiments conducted on NAS-Bench-201 and TransNAS-Bench-101 demonstrate the superiority of our proposed cross-task search schemes, namely ArchLock-TU and ArchLock-TK in transferability reduction compared to state-of-the-art source-only NAS methods.

## 7 ACKNOWLEDGEMENT

This work is supported in part by the U.S. National Science Foundation under Grants OAC-2319962, CNS-2239672, CNS-2153690, CNS-2326597, and CNS-2247892.

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

# A    DERIVATION IN TASK EMBEDDING

Given the source task embedding $\mathbf{z}_{\mathcal{S}}$, we first generate a vector $e_i$ that is orthogonal to $\mathbf{z}_{\mathcal{S}}$, which can be obtained via Gram–Schmidt process.

To satisfy the desired cosine similarity $d$, i.e.,

$$S_{cos}(\mathbf{z}_{\mathcal{S}}, \mathbf{z}_{\mathcal{T}}) := cos(\theta) = d, \tag{6}$$

where we use $S_{cos}$ to denote cosine similarity and $\theta$ is corresponding angle. Here we set the norm of the target task embedding the same as $\mathbf{z}_{\mathcal{S}}$, so we will have:

$$\mathbf{z}_{\mathcal{T}_i} = cos(\theta) \cdot \mathbf{z}_{\mathcal{S}} + \sin(\theta) \cdot e_i \cdot ||\mathbf{z}_{\mathcal{S}}||, \tag{7}$$

According to Eq. (6), we also have $\theta = \arccos(d)$. Therefore. Eq. (7) can be expressed as:

$$\mathbf{z}_{\mathcal{T}_i} = d \cdot \mathbf{z}_{\mathcal{S}} + \sin(\arccos(d)) \cdot e_i \cdot ||\mathbf{z}_{\mathcal{S}}||. \tag{8}$$

# B    DETAILS OF ZERO-PROXY

In ArchLock, we utilize zero-cost proxies (ZCPs) obtained from the NAS-Bench-Suite-Zero tool Ruchte et al. (2020). This tool assesses the performance of various ZCPs, and we specifically focus on data-dependent ZCPs since we believe that architectural performance could vary depending on the data involved. The data-dependent ZCPs consider data/tasks to compute scores but do not update the gradient for weight optimization.

The NAS-Bench-Suite-Zero tool provides a total of 9 data-dependent ZCPs. We select 7 out of these 9, namely fisher Turner et al. (2020), flops Ning et al. (2021), grad-norm Abdelfattah et al. (2021), grasp Wang et al. (2020), jacov Mellor et al. (2021), nwot Mellor et al. (2021), and snip Lee et al. (2019b). Two ZCPs, namely epe-nas Lopes et al. (2021) and plain Ning et al. (2021), are excluded from our selection.

We exclude epe-nas because it is designed specifically for the classification task. As for plain, it exhibits the lowest Spearman rank correlation coefficient with validation accuracies among the 9 ZCPs. Furthermore, extensive experiments conducted in Krishnakumar et al. (2022) demonstrate that including more than 6-8 ZCPs only leads to marginal improvements. Therefore, we consider the inclusion of 7 ZCPs to be sufficient for our purposes.

The technique of each ZCP is briefly described below:

- fisher: It computes the sum of gradients of the activations in the network.

- flops: It measures the number of floating-point operations (flops) required to process an input through the network. It provides an estimate of computational complexity.

- grad-norm: It calculates the sum of Euclidean norms of gradients.

- grasp: It approximates the change in gradient norm.

- jacov: It measures the covariance of prediction Jacobian matrices across a minibatch of data. It captures the relationship between predictions and their gradients.

- nwot: It examines the overlap of activations between data in a mini-batch for an untrained network.

- :snip: It estimates the change in the loss function.

It is important to note that while not all of these metrics were initially introduced as ZCPs, they have been later utilized in the field of NAS to assess architecture performance without the need for extensive training. These metrics have been demonstrated effective in providing reliable estimations of performance.

## C  IMPLEMENTATION OF FUNCTIONS IN CROSS-TASK SEARCH

---

**Algorithm 2** Binary Predictor with Sorting Algorithm

---

1: **function** BPWSORT($popu, P, \mathbf{z}_{\mathcal{S}}, \mathbf{z}_{\mathcal{T}_i}$)
2:     Zero-initialize matrix $comp_s$ and $comp_t$ of shape $[len(popu), len(popu)]$
3:     **for** $a, b \in range(len(popu))$ with $b > a$ **do**
4:         Encode $arch_a, arch_b \rightarrow arch\_pair$
5:         $p_a, p_b \leftarrow P(arch\_pair, \mathbf{z}_{\mathcal{S}})$
6:         **if** $p_a > p_b$ **then**
7:             $comp\_s[a, b] \leftarrow 1$
8:         **else**
9:             $comp\_s[b, a] \leftarrow 1$
10:        **end if**
11:        **for** each $\mathbf{z}_{\mathcal{T}_i}$ **do**
12:            $p_a, p_b \leftarrow P(arch\_pair, \mathbf{z}_{\mathcal{T}_i})$
13:            **if** $p_a > p_b$ **then**
14:                $comp\_t[a, b] \mathrel{+}= 1$
15:            **else**
16:                $comp\_t[b, a] \mathrel{+}= 1$
17:            **end if**
18:        **end for**
19:    **end for**
20:    $sum_s \leftarrow$ Sum $comp_s$ along dimension 0
21:    $\mathcal{S}_{rank} \leftarrow$ Get the index of elements of $sum_s$ in descending order
22:    $sum_t \leftarrow$ Sum $comp_t$ along dimension 0
23:    $\mathcal{T}_{rank} \leftarrow$ Get the index of elements of $sum_t$ in descending order
24:    **return** $\mathcal{S}_{rank}, \mathcal{T}_{rank}$
25: **end function**

---

**Algorithm 3** Update History in cross-task search

---

1: **procedure** UPDATEHISTORY($history, arch, P, \mathbf{z}_{\mathcal{S}}, \mathbf{z}_{\mathcal{T}_i}$)
2:     $history$.append($arch$)
3:     **if** $len(history) > k$ **then**
4:         $\mathcal{S}_{rank}, \mathcal{T}_{rank} \leftarrow$ BPwSort($history, P, \mathbf{z}_{\mathcal{S}}, \mathbf{z}_{\mathcal{T}_i}$)
5:         Calculate fitness score f for each $arch \in history$
6:         $history \leftarrow$ top-$k$ $arch$s with highest f in $history$
7:     **end if**
8: **end procedure**

---

## D  NAS-BENCH-201

The performance of different task pairs in NAS-Bench-201 can be observed in Fig. 4. From the figure, it is evident that there is a relatively high correlation between CIFAR-10 and CIFAR-100, indicating that architectures that perform well on CIFAR-10 tend to also perform well on CIFAR-100. However, the correlation between CIFAR-100 and ImageNet-16-120 is lower. Indeed, the absence of a perfect positive correlation between task pairs in NAS-Bench-201 suggests that the architectural choices yielding optimal performance on one task may not translate to the best performance on another task. This observation indicates that there is room for exploration and development of secure architectures that can effectively defend against malicious transfer attacks.

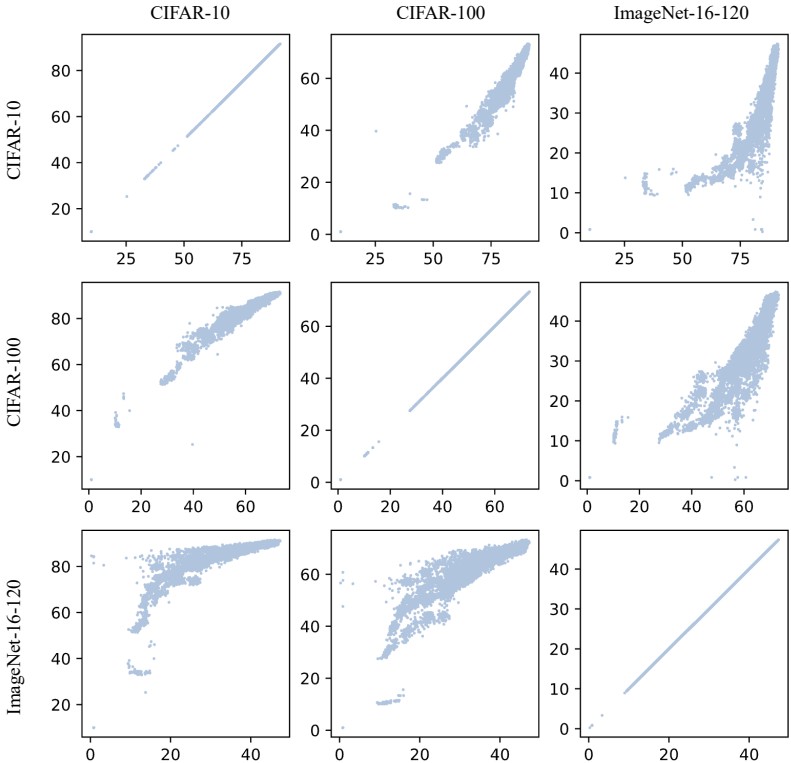

Figure 4: The correlation of each task pair in NAS-Bench-201.

# E  TRANSNAS-BENCH-101

The correlation of each task pair in TransNAS-Bench-101 is shown in Fig. 5, where Acc. denotes accuracy and Neg. denotes negative. Here we use the negative loss for RL to be consistent with other metrics, i.e., the higher value indicates better performance. Compared to NAS-Bench-101, the task pair in this benchmark shows a relatively lower correlation, which indicates the transferability of an architecture varies when the task changes. By leveraging the understanding that architectures may perform differently across various tasks, we can focus on designing models that only excel in their source tasks but also possess robustness and resilience against unauthorized transfer. This opens up opportunities for research and innovation in creating architectures that prioritize security and can withstand attempts to extract and transfer their knowledge maliciously.

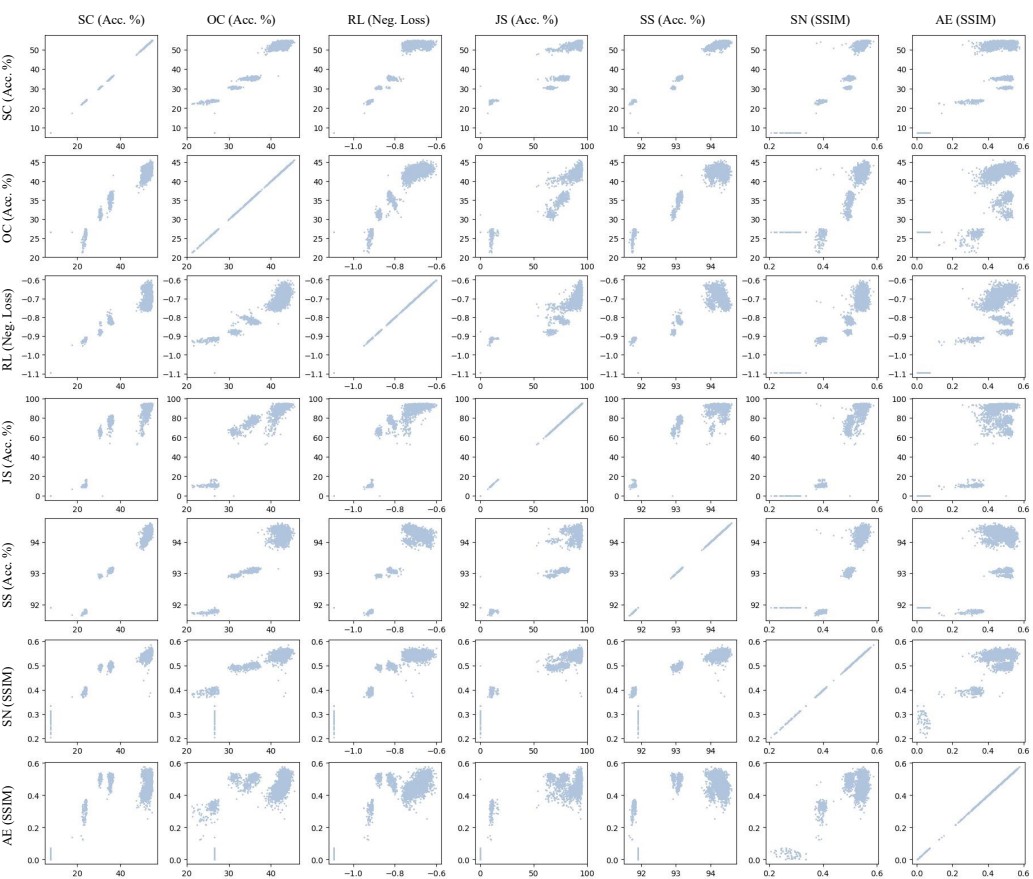

Figure 5: The correlation of each task pair in TransNAS-Bench-101.

