# OpenReview forum: "ArchLock: Locking DNN Transferability at the Architecture Level with a Zero-Cost Binary Predictor"
_ICLR.cc/2024/Conference — ICLR 2024 poster_

### Official Review · Reviewer_boEZ · 2023-10-30

**Soundness:** 2 fair
**Presentation:** 3 good
**Contribution:** 2 fair
**Rating:** 3
**Confidence:** 5

**Summary:**

This paper presents a methodology to degrade NN performance on new tasks. Specifically, the paper mentions that adversaries may want to adapt a pretrained NN to a new task while violating its terms of use. To mitigate this issue, the presented methods performs a form of neural architecture search to find NN architectures that degrade performance on the tasks for which the NN was not trained.

**Strengths:**

- Demonstrates results on transnasbench.
- Leverages zero-cost proxies in a new way.

**Weaknesses:**

While this paper is interesting, I am not quite convinced of the motivation behind it. If you want to prevent others from fine-tuning a NN, why even release its parameters to begin with? perhaps in this use-case, the model should not even be released? Also, the definition of "task" is very broad. Would additional data be considered a new task? for example, more 32x32 images to be classified into cifar 10 classes for a cifar-10 NN? Or is it just when the classification head is modified?

Other weaknesses include:
- a limited evaluation on NAS benchmarks, making it harder to appreciate the motivation of the paper. If evaluation was done on some NN that someone wants to protect, then it would've helped.
- 2% performance degradation on CIFAR-leve NNs is actually quite large.
- Zero-cost proxies are meant to guage accuracy in general and have not been verified to work for this  task of minimizing out-of-training-distribution accuracy.

**Questions:**

Can you please provide responses to the weaknesses above. The work is interesting but not fully convincing in its current form.

---

> ### Author Response · Authors · 2023-11-14
> **Motivation**
>
> Thanks for your question and valuable input. However, we think there is a **misunderstanding about our work** and we would like to clarify below.
>
> Our objective is **not to degrade NN performance on new tasks** as you stated in Summary. Instead, our focus is to find model architectures that perform well on their intended tasks while exhibiting low transferability, or degraded performance, on unauthorized target tasks.
>
> **[Motivation]**
>
> The motivation for our approach **stems from ethical considerations and the potential misuse of machine learning models**, as highlighted in recent works such as [1,2,3,4].  For example, a recent work [1] has demonstrated the open-sourced models, including Stable Diffusion, Latent Diffusion, DALL·E 2-demo, and DALL·E mini can be misused to generate unsafe images and hateful memes. Also, another recent work has revealed that fine-tuning aligned language models (e.g., Llama-2) compromises safety [2].
> Thus, model owners are willing to release their models to contribute to the research community while still seeking to control and limit their usage to prevent potential misuse or abuse.
>
> Our proposed method, which reduces transferability, can thus serve as a tool to safeguard the intellectual property (IP) of the model. It **provides a way to protect against unauthorized use, particularly in scenarios where there may be ethical or safety concerns associated with the application of the model on certain tasks**. We will clarify our motivation in the final version.
>
> ---
> Reference:
>
> [1] Unsafe Diffusion: On the Generation of Unsafe Images and Hateful Memes From Text-To-Image Models, ACM SIGSAC Conference
> on Computer and Communications Security (CCS). ACM, 2023.
>
> [2] Fine-tuning aligned language models compromises safety, even when users do not intend to! arXiv preprint, arXiv preprint arXiv:2310.03693, 2023.
>
> [3] Self-destructing models: Increasing the costs of harmful dual uses of foundation models. In Proceedings of the 2023 AAAI/ACM
> Conference on AI, Ethics, and Society, pages 287–296, 2023.
>
> [4]  Identifying and mitigating the security risks of generative ai. arXiv preprint arXiv:2308.14840, 2023.

---

> ### Author Response · Authors · 2023-11-14
> **Address Weaknesses**
>
> **[Task Definition]**
>
> In our work, we did not consider adding more 32x32 images into CIFAR-10 (e.g., via augmentation) for classification as constituting a new task. In the supervised learning setting, a task is defined by both its domain space (data distribution, e.g., CIFAR-10 vs. ImageNet) and its label space (e.g., classification with different classes, classification vs. object detection). Therefore, a task is considered a new task when either its domain space is different or its label space is different.
>
> **[Evaluation and Performance]**
>
> We would like to emphasize that our objective in this paper is **not centered on protecting specific neural networks (NNs)**. Instead, we aim to discover secure architectures for individuals who seek to build NNs for various tasks. The obtained secure architecture can then be deployed with reduced risk of misuse.
>
> Regarding performance degradation, it is the "security budget" for transferability reduction, which is simply a design choice that is faced by any security solutions, i.e., the tradeoff between security and performance. As a generic solution, our method can adjust this budget by modifying $\lambda$ in the fitness score. A higher $\lambda$ will result in lower performance degradation on the source task.
>
> **[Zero-cost Proxies]**
>
> It appears there might be **a misunderstanding regarding our use of zero-cost proxies**. In our work, we employed them to train a performance predictor. For instance, some studies use architectures' accuracy to train NAS predictors [1], whereas we leverage zero-cost proxies. To tackle out-of-training-distribution challenges, the predictor undergoes meta-training. This approach aligns with the rationale presented in [2], where meta-learning has been demonstrated to be effective in out-of-training-distribution scenarios [3].
>
> ---
> Reference:
>
> [1] Brp-nas: Prediction-based nas using gcns. Advances in Neural Information Processing Systems, 33:10480–10490, 2020.
>
> [2]  Rapid neural architecture search by learning to generate graphs from datasets. In International Conference on Learning Representations, 2021.
>
> [3] Model-agnostic meta-learning for fast adaptation of deep networks. In International conference on machine learning, pages 1126–1135. PMLR, 2017.

---

> ### Author Response · Authors · 2023-11-14
> **Kind Request**
>
> Considering you are the **sole reviewer who has provided a negative rating**, we value your feedback and would like to ensure that all your concerns are addressed. We sincerely appreciate your acknowledgment that our work is interesting. **Our work represents a pioneering effort in providing a defense against unauthorized transfer at the architectural level, and the compelling results obtained underscore the effectiveness of our approach.** We kindly request your reconsideration of the rating, taking into account the clarification and context provided.

---

> ### Author Response · Authors · 2023-11-19
> **Looking forward to your response**
>
> We would like to emphasize the practical significance of our contribution to the field. The problem we tackled holds substantial practical relevance, and our motivation is thoroughly justified. We understand the importance of a fair evaluation and believe that the current rating may underestimate the true value of our work. We hope that you can consider raising the rating score based on the information we provided.

---

> > ### Comment · Reviewer_boEZ · 2023-11-21
> > **Response to the authors' clarifications**
> >
> > Thank you for responding to my concerns regarding the motivation of your work. The stable diffusion example that you gave makes sense and I do have a better understanding of the work's purpose thanks to your response. I do hope that this makes it to your updated manuscript.
> >
> > However, there are still some issues:
> > 1- If image generation and diffusion models are the key motivating example, why not try your method on this class of models instead of the NB201 and TransNB evaluations?
> > 2- I still maintain that a 2% accuracy degradation is too high on a CIFAR-level task.
> >
> > Not that I do understand that you want to "degrade performance on unauthorized target tasks" - this is what I meant by "new tasks" in my summary.
> >
> > The presented work performs NAS with an additional objective of _low_ accuracy on target or unseen tasks. For the case that the target task is known, directly comparing to conventional multiobjective NAS (e.g. table 3) with an additional objective of degrading accuracy on target task would make sense. This would make for a more fair comparison to Archlock-TK.
> >
> > I'm reluctant to increase my score because of the concerns above. If there was a 4 rating, perhaps I would increase to that. The work in its current form has missing comparisons, and unrealistic evaluation only on NAS benchmarks which may not necessarily be transferrable to the real-world scenarios described by the authors.

---

> ### Author Response · Authors · 2023-11-21
>
> Thanks for your reply.
>
> **[For issue 1]**
>
> We used image generation and diffusion models as examples to address your question, "Why even release its parameters to begin with?" and to provide a better understanding. However, the vulnerability is general and not limited to generative tasks. Also, like all other work conducted on NAS Benchmarks, the objective is to provide a fair comparison. Your current concern seems centered on the existing NAS Benchmarks being less meaningful (or are you questioning the practical significance of NAS?). However, it's crucial to note that constructing NAS benchmarks is not the primary focus of our work. If someone proposes a diffusion model-based benchmark for generative tasks, we are willing to test our method on that. Additionally, our method holds promise for real-world scenarios as NAS has already been employed to design high-performance models for tasks. With our approach, the resulting models can not only exhibit superior performance but also boast enhanced security measures.
>
> **[For issue 2]**
>
> We have explained that accuracy degradation is a design choice, and one can adjust it by modifying
> λ in the fitness score. We would like to provide more experiments as a discussion in the appendix.
>
> **[For the newly raised question: comparison for TK]**
>
> As we mentioned in the introduction, our focus is on how to determine and simulate the potential target task if it is not specified, and our method centers on this case and proposes TU. **Archlock-TK is already a comparison method designed for TU, and is not meaningful to include other comparison methods for our comparison method.** Also, we have explained why current cross-task or multi-objective NAS cannot be compared with TU in Section 2.1.
>
> **[Summary]**
>
> **Our formulation and methodology constitute the primary contribution we aim to make to the field, which has been acknowledged by other reviewers who think that our formulation is solid.**  We are the first to address this vulnerability by proposing a methodology leveraging NAS, thus we experiment with NAS benchmarks. Since more advanced benchmarks have not been proposed,  the experiment design you suggested is challenging at this time. However, our current experiments are sufficient to demonstrate the effectiveness of our method, and it not undermine our contribution. It's crucial to emphasize that our motivation is well-justified, the vulnerability is practical, and our methodology is well-formulated.
>
> Moreover, **considering that we were the first to identify this vulnerability and propose an effective mitigation method, our work has the potential to raise awareness among other researchers regarding this vulnerability. This could pave the way for them to propose more advanced solutions, building upon the foundation laid by our work.**
>
> **If you also agree that a rating of 3 is underrated, we kindly encourage you to raise it to the next level.**

---

### Official Review · Reviewer_umc9 · 2023-11-01

**Soundness:** 3 good
**Presentation:** 3 good
**Contribution:** 3 good
**Rating:** 8
**Confidence:** 4

**Summary:**

This paper presents a cross-task NAS framework to find an architecture to mitigate unauthorized DNN transferability.  A binary predictor using multiple zero-cost proxies is proposed to accelerate the NAS procedure. The results and the ablations demonstrate the effectiveness of the proposed method.

**Strengths:**

1. The whole formulation of reducing transferability at the architecture level involved with architecture search is solid.
2. The proposed nas method based on binary predictor is efficient and effective in designing model architectures with low transferability.

**Weaknesses:**

1. Since the proposed method is based on a predictor, maybe it is better to cite a series of predictor-based NAS work. For example, PINAT: A Permutation INvariance Augmented Transformer for NAS Predictor AAAI 2023 TNASP: A Transformer-based NAS Predictor with a Self-evolution Framework NeurIPS 2021 and so on.
2. It seems that there is no training details about the binary predictor, What is the training cost for this predictor? Is one pre-trained predictor suitable for processing the architectures from different search spaces?

**Questions:**

NA

---

> ### Author Response · Authors · 2023-11-14
>
> Thanks for pointing out these references; we will cite these works in the proper place in the final version.
>
> **[Training details]**
>
> The training process of our binary predictor has been provided in Section 4.3. Specifically, the predictor is constructed using a four-layer multi-layer perceptron (MLP). We also conducted experiments with various designs, including different architecture parameters and optimizers. Some of the experiment records are presented in the table below. The selected design, highlighted in bold, achieved an accuracy of 87% with a training cost of approximately 5 GPU hours. We would like to include this information as supplemental material in the final version.
>
>
> | # of Layers | Layer Configuration | Optimizer | Accuracy |
> |:-------------:|:---------------------------:|:-----------:|:----------:|
> | 3           | [256, 64, 32]        | SGD       | 85%      |
> | 4           | [256, 64, 32, 16]    | SGD       | 83%      |
> | 4           | [256, 64, 32, 16]    | Adam      | 87%      |
> | **4**       | **[128, 64, 32, 16]**   | **Adam**     | **88%**     |
> |5	|[128, 64, 32, 16, 8]	|Adam	|87%|
>
> The experiments in the above table share common settings, which are as follows:
>
> - Loss function: BCEWithLogitsLoss.
> - Learning rate: set to 0.05.
> - Learning rate scheduler: StepLR with a step_size of 5 and a gamma of 0.5.
>
> Since the input of the pre-trained predictor consists of the architecture embedding (see Fig. 1), once the search space is defined differently (e.g., DARTS vs NAS-Bench-201), the pre-trained predictor would not fit anymore. It is worth noting that it is common to train a new predictor for different search spaces, including the two papers you mentioned.
> However, given that the cell-level search space of TransNAS-Bench-101 is a subset of NAS-Bench-201, our pre-trained predictor can be applied to both.
>
> Given that **training detail is the only concern raised**, and considering **your acknowledgment that our formulation is solid and the proposed framework is efficient and effective, we kindly request that you reconsider your rating for this work**. We believe the additional information provided addresses the concern and contributes to a more thorough understanding of our approach. We appreciate your time and consideration.

---

> ### Author Response · Authors · 2023-11-19
>
> We would appreciate confirmation that we have adequately addressed all your concerns. Thank you once again for your time.

---

> > ### Author Response · Authors · 2023-11-22
> >
> > We look forward to your reply before the rebuttal period ends tomorrow. Thanks!

---

### Official Review · Reviewer_ySE1 · 2023-11-06

**Soundness:** 4 excellent
**Presentation:** 2 fair
**Contribution:** 3 good
**Rating:** 6
**Confidence:** 4

**Summary:**

This paper prposes ArchLok to mitigate unauthorizaed DNN transfer. ArchLock first encodes the NN architecture to evaluate rank two architectures with task embeddings, then perform neural architecture search to find archs that are good on source tasks but bad on target tasks. Evaluation on NAS-Bench-201 and Trans-Bench-101 demonstrate that ARchLock significantly reduces the transferbility.

**Strengths:**

By addressing security at the architecture level, ArchLock potentially fills a gap left by other security measures that focus on the model parameter level. This provides a more holistic defense strategy for DNN models.

ArchLock focuses more on the architecture rankings rather than the actual performance numbers, and utilize efficient zero-cost proxies as supervision. This approach can be scaled to any size of architecture pool and reduces the cost of training several architectures from scratch.

Experiments on NAS-Bench-201 and TransNAS-Bench-101 demonstrate the effectiveness of ArchLock. It can effectively degrade the performance on target tasks by up to 30% while preserving the performance on source tasks.

**Weaknesses:**

The details of S / TU / TK are not clearly described. Algorithm 1 shows the cross-task search when the target task is known, but it does not discuss how the other two baselines are performed. Additionally, it is still unclear how the GraphEncoder (Figure.1) is executed and how task embedding is extracted. How much overhead does  task embedding take for each new task?

Are the numbers in Tab 1 and 2 real measurements, or are they directly taken from NAS-Bench/TransNAS-Bench? The zero-cost proxies/predictors are trained on the same set of datasets, which may lead to potential overfitting. Evaluating on unseen and large-scale datasets (e.g., ImageNet [can be a subset with ful 224x224 resolution], Miniplaces) is necessary to demonstrate effectiveness.

ArchLock aims to design architectures that show less transferability on new tasks, but it does not discuss what kind of architecture leads to poor transferability. For example, do different datasets dislike different architecture designs, or there is an type of architecture that transfers bad generally on all tasks? It would be beneficial to provide visualizations and discussions of general un-transferable architectures so that further work can gain inspiration and insights.

**Questions:**

See weakness above

---

> ### Author Response · Authors · 2023-11-14
>
> Thanks for your valuable comments. We would like to address your concerns one by one.
>
> **[Details of S/TU/TK]**
>
> For S (source-only search scheme), it indeed follows the common NAS search scheme and does not consider the target task during its search process. Specifically, instead of using Eq. (5), its fitness score can simply be $f=-\\mathcal{S}_{rank}$, where higher ranks correspond to lower values, resulting in a higher fitness score.
>
> Moreover, Algorithm 1 indeed encapsulates the cross-task search process for both TU and TK, while the difference between them can be better understood with details in Line 11-18 of Algorithm 2 in Appendix C. Specifically, as mentioned in Section 4.2, the target task is known for TK, so it only has one $\\mathcal{T}$ and goes through one loop in Lines 11-18 with $z_\\mathcal{T}$ extracted from the known/true target task. In contrast, for TU with no knowledge of the true target task, we simulated 10 $z_{\\mathcal{T}_i}$ in our experiments following the method described in Section 3.1. Thus, TU has to go through 10 loops according to Lines 11-18 in Algorithm 2. After the search terminated, we then examined the performance of the searched architecture on the true target task, which remained unknown during the search. This is a more challenging defense scenario, but TU still strikes a better balance between the performance on the source task and transfer vulnerability compared to S.
>
> **[Graph Encoder]**
>
> The graph encoder converts the directed acyclic graph of the architecture cell into an adjacency matrix, which will be flattened into a 1-D array as the architecture embedding.
>
> For example, in NAS-Bench-201, each architecture consists of a predefined skeleton with a stack of the searched cell (see Fig 1 in [1]). Architecture search is thus transformed into the problem of finding a good cell. Each cell is represented as a directed acyclic graph with 6 edges, each associated with an operation selected from a predefined set (i.e., [zeroize, skip-connect, 1x1 conv, 3x3 conv, 3x3 avg pool]). To illustrate, consider the following cell configuration:
>
> - node-0: the input tensor
> - node-1: conv-3x3( node-0 )
> - node-2: conv-3x3( node-0 ) + avg-pool-3x3( node-1 )
> - node-3: skip-connect( node-0 ) + conv-3x3( node-1 ) + skip-connect( node-2 )
>
> We apply one-hot encoding to encode the cell, and get its adjacency matrix (with the row indicating the edge and the column indicating the operation):
>
> $$
> \\left(\\begin{array}{ccccc}
>  0 & 0 & 0 & 1 & 0 \\\\
>   0 & 0 & 0 & 1 & 0 \\\\
>   0 & 0 & 0 & 0 & 1 \\\\
>   0 & 1 & 0 & 0 & 0 \\\\
>   0 & 0 & 0 & 1 & 0 \\\\
>   0 & 1 & 0 & 0 & 0 \\\\
> \\end{array}\\right) $$
>
> Thus, our graph encoder can convert an architecture into a 6x5 matrix, which is then flattened into a 1-D array of size 30 as the architecture embedding.
>
> **[Task Embedding]**
>
> Regarding task embedding, we followed the method outlined in [2] as we mentioned in the paper (we refer readers to Section 3.1 in [2] for more details), with its implementation available at [this link](https://github.com/awslabs/aws-cv-task2vec). In our implementation, we used ResNet50 pre-trained on ImageNet as a feature extractor, and the overhead for computing a task embedding mainly comes from training the "head" layer of the given task. Given the pre-trained feature extractor already provides rich representation, only re-training the head can converge fast and the whole process can be done efficiently.
>
>
> ---
> Reference:
>
> [1] Nas-bench-201: Extending the scope of reproducible neural architecture search. In International Conference on Learning Representations (ICLR), 2020.
>
> [2] Task2vec: Task embedding for meta-learning. In Proceedings of the IEEE/CVF international conference on computer vision, pages 6430–6439, 2019.

---

> ### Author Response · Authors · 2023-11-14
>
> **[Measurement and Overfitting]**
>
> The reported numbers in Tables 1 and 2 are indeed derived from the benchmark to ensure a fair comparison by avoiding the influence of hyperparameter tuning.
>
> Regarding the zero-cost predictor, we would like to emphasize that it is not trained on the same set of datasets. Instead, it is meta-trained on ImageNet, as explicitly mentioned in Section 4.3. This intentional separation from the datasets used for the main evaluation mitigates concerns about overfitting, as the predictor is exposed to a diverse range of data during meta-training. We believe this design choice strengthens the generalization capabilities of our approach.
>
> **[Transferability Discussion]**
>
> Based on the observation from our experiments, there is not a type of architecture that transfers badly for all tasks. However, we appreciate your constructive suggestion, which brings up an interesting direction that is worth exploring in the future.
>
> ---
> We hope we have addressed all your concerns. Please let us know if you have any further questions. **If not, we kindly request that you reconsider the rating, considering your acknowledgment of the novelty of our work and its significance as a holistic defense strategy for DNN models.** Thanks for your time!

---

> ### Author Response · Authors · 2023-11-19
> **Any remaining concerns?**
>
> We are looking forward to your feedback and wondering if any concerns remain.

---

> > ### Comment · Reviewer_ySE1 · 2023-11-21
> >
> > Thanks for the authors' reponse partially addressed my concerns.  The questions about S/TU/TK, GraphEncoder, Task Embeddings now are clear to me. However, I am still holding concerns about transferability and general patterns for less-transferable architecture.
> >
> > I will keep my current reviews.

---

### Author Response · Authors · 2023-11-21
**To All Reviewers**

As the rebuttal period is drawing to a close, we sincerely hope that we have adequately addressed all your concerns. We would greatly appreciate your confirmation in this regard. Furthermore, if you choose to maintain your current ratings after we provide additional information or clarification, we kindly request justification for your decision. Your time and consideration are highly valued. Thank you.

---

### Meta-Review · Area_Chair_pfu1 · 2023-12-05

**Metareview:**

This paper proposes a new formulation of finding a neural architecture that is strong on one task but transfers poorly to others.
This problem setting was criticized as unintuitive by Reviewer boEZ, since it is unclear in which circumstances this would be a desideratum of a vision model. The reviewers gave as an example that a diffusion model should not be good at generating hateful memes, which convinced the reviewer as a strong example. While there are no experiments on diffusion models, this is a good justification why the formulation might be useful in the future.
The method is fairly standard meta-learning. The results are not stellar, with even the most positive reviewer stating that 2% higher errors on CIFAR-10 are too much. One issue in NAS research has been reproducibility, and while some code is contained in the supplementary material zip file, this is not mentioned in the paper or appendix. I strongly ecourage the authors to make this code available to the public; I would like to emphasize that this is crucial to lay a foundation for follow-up work. The fact that the paper is based on top of standard implementations of ZC proxies and benchmarks in NASlib speaks in favour of reproducibility.
The most positive review was rather high-level and asked the authors to cite two additional papers by the same authors (also, at first glance these are not the most immediately related works out there; I would advise against giving into this request); this cannot be held against the authors, but I am not weighing that positive review highly.
Overall, the paper is borderline. I do like the aspect of introducing a new type of robustness criterion and that code is attached. I trust that it will be made code publicly available (on github or alike).

**Justification For Why Not Higher Score:**

Standard methodology, unclear applicability.

**Justification For Why Not Lower Score:**

Introduction of a new robustness criterion that architectures might want to be optimized for.

---

### Decision · Program_Chairs · 2024-01-16

Accept (poster)